# Intercropping Okra and Castor Bean Reduces Recruitment of Oriental Fruit Moth, *Grapholita molesta* (Lepidoptera: Tortricidae) in a Pear Orchard

**DOI:** 10.3390/insects14110885

**Published:** 2023-11-16

**Authors:** Zhen Li, Jianmei Yu, Haoyang Xu, J. P. Michaud, Yanjun Liu, Xiaoxia Liu, Huanli Xu

**Affiliations:** 1Department of Entomology and MOA Key Lab of Pest Monitoring and Green Management, College of Plant Protection, China Agricultural University, Beijing 100193, China; lizhencau@cau.edu.cn (Z.L.); cauyjm@163.com (J.Y.); xhy797911@163.com (H.X.); liuyanjun@ioz.ac.cn (Y.L.); liuxiaoxia611@cau.edu.cn (X.L.); 2Shandong Zibo Academy of Agricultural Sciences, Zibo 255033, China; 3Department of Entomology, Kansas State University, Agricultural Research Center-Hays, Hays, KS 67601, USA; jpmi@ksu.edu

**Keywords:** cinnamaldehyde, dibutyl phthalate, integrated pest management, repellents, thymol, *Trichogramma dendrolimi*

## Abstract

**Simple Summary:**

This study used field observations to demonstrate that intercropping okra in a pear orchard reduced pheromone trap catches of *Grapholita molesta* in two years, whereas intercropping with castor bean reduced them in one year, relative to natural grass cover. GC-MS assays showed that hydrocarbons, phenols, and ketones predominated in the volatiles emitted from okra, whereas aldehydes, ketones, and esters were the most abundant volatiles emitted from castor bean. Five of these compounds exhibited repellency to *G. molesta* in olfactometry assays, especially cinnamaldehyde, dibutyl phthalate, and thymol; the former also served as an attractant for the egg parasitoid *Trichogamma dendrolimi*, which is often used in augmentation biocontrol of the moth.

**Abstract:**

Intercrops can lower pest densities by increasing plant diversity, altering chemical communication in the arthropod community, and integrating well with other IPM tactics. We used two years of field observations and Y-tube olfactometer assays to explore the effects of intercropping a pear orchard with okra and castor bean on the cosmopolitan fruit-boring pest *Grapholita molesta* (Lepidoptera: Tortricidae). Intercropping okra reduced *G. molesta* trap catches in the pear orchard in both years, and intercropping with castor bean reduced them in the second year. Hydrocarbons, phenols, and ketones predominated in the GC-MS assay of okra volatiles, whereas castor bean volatiles were rich in aldehydes, ketones, and esters. Five of the commercially available volatiles released by these plants exhibited repellency to *G. molesta* in olfactometer trials, especially cinnamaldehyde, dibutyl phthalate, and thymol; the former compound also exhibited attraction to the egg parasitoid *Trichogamma dendrolimi* (Hymenoptera: Trichogrammatidae). In addition to their repellent properties, okra and castor bean may enhance integrated control of *G. molesta* in orchards by hosting prey that support populations of generalist predators that either provide biological pest control services within the orchard ecosystem or generate non-consumptive effects that contribute to pest deterence. Among the plant volatiles evaluated, cinnamaldehyde has the best potential for deployment in orchards to repel *G. molesta* without disrupting augmentative releases of *T. dendrolimi*.

## 1. Introduction 

An effective integrated pest management (IPM) strategy should harness natural ecological mechanisms to maintain pests below their economic thresholds in a cropping system using tactics that are cost-effective relative to alternative control measures [1,2,3]. Intercropping is a cultural control tactic dating back to antiquity that has recently drawn renewed attention for its potential to deliver multiple ecological benefits, including increased biodiversity and enhanced chemical communication among plants, pests, and natural enemies [4,5,6]. Intercropping has been most thoroughly studied in annual crops, yielding management benefits that include repulsion of crop pests, attraction and retention of natural enemies that enhance pest suppression, and improved plant resistance via nutritional enhancement and water retention [7]. In comparison, perennial orchards provide a more temporally stable habitat that is intrinsically more amenable to natural pest control, unless management practices disrupt these services [8]. Intercropping orchards with flowers and grasses, whether wild or artificially sown, has been shown to promote functional diversity and natural enemy residence, often supplementing other IPM tactics [9]. In addition, any value the intercrop might have as either food or forage will reduce the cost of its deployment. Nevertheless, research is limited to intercropping cash crops in orchards for pest control purposes, and the potential functions of their semiochemicals in mediating ecological communications among plants, pests, and natural enemies remain largely unexplored in horticultural agroecosystems. 

The oriental fruit moth, *Grapholita molesta* (Busck) (Lepidoptera: Tortricidae), is a cosmopolitan pest of rosaceous fruit trees in temperate climates [10]. In recent years, losses in fruit production due to damage by *G. molesta* have worsened as global warming drives changes in thermal regimes and cultural practices [11]. Control of *G. molesta* is challenging because larvae feed internally within plant parts from eclosion until pupation; there are 3–6 overlapping generations per season; and many populations have evolved resistance to commonly used insecticides [12,13]. Current management of *G. molesta* in fruit orchards relies on a combination of insecticide applications targeting adult moths, mating disruption with sex pheromones, and inundative releases of *Trichogramma* spp. egg parasitoids (Hymenoptera: Trichogrammatidae) during peak oviposition periods [14,15]. However, IPM programs for *G. molesta* would benefit from the integration of novel control tactics, given the diminishing efficiency of mating disruption [16], its evolution of resistance to organophosphate and carbamate insecticides [12,13], and the susceptibility of egg parasitoids to environmental factors [17,18]. Intercropping orchards with cash crops have the potential to enhance the IPM of *G. molesta* while providing off-setting economic returns. 

Okra is the fruit of a plant in the mallow family (Malvaceae), *Abelmoschus* (*Hibiscus*) *esculentus* (L.) Moench, which has a high nutritional value for human consumption. It has economic value as a vegetable crop worldwide, especially in eastern Europe, Africa, and Asia [19,20]. Okra is the source of more than a hundred phytochemicals, including many with pharmacological and toxicological uses and pest-resistant properties [20]. For example, extracts of *Hibiscus abelmoschus* L. have demonstrated larvicidal activity against mosquitoes [21]. 

Castor bean, *Ricinus communis* L. (Euphorbiaceae), is an important non-edible oilseed crop of value to the manufacturing industry [22], and its oil contains many bioactive compounds with a wide range of pharmaceutical properties [23]. Castor oil exhibits direct toxicity to the pickleworm, *Diaphania nitidalis* (Stoll) (Lepidoptera: Crambidae) [24], the diamondback moth, *Plutella xylostella* L. (Lepidoptera: Plutellidae) [25], and also inhibits reproduction in the cattle tick, *Rhipicephalus microplus* [26]. Volatiles from flowering *R. communis* plants are repellent to the whitefly *Bemisia tabaci* (Gennadius) (Hemiptera: Aleyrodidae) [27], and leaf extracts have shown insecticidal and oviposition deterrent activity against the bruchid beetle *Callosobruchus chinensis* (L.) (Coleoptera: Bruchidae) [28] and toxicity to the yellow sugarcane aphid, *Sipha flava* (Forbes) (Hemiptera: Aphididae) [29]. Here we explored the potential benefits of intercropping okra and castor bean in a pear orchard for integrated pest management of *G. molesta*. 

The objectives of the present study were: (1) to compare the effects of okra and castor bean intercrops, relative to grass cover, on *G. molesta* abundance; (2) to analyze the profile of volatile organic compounds released by okra and castor bean plants; and (3) to test the responses of *G. molesta* adults and its egg parasitoid, *T. dendrolimi*, to okra and castor bean volatiles in olfactory bioassays. The results were expected to identify the potential benefits of these plants for IPM of *G. molesta* in an orchard environment.

## 2. Materials and Methods

### 2.1. Intercrop Experiment 

The experiment was conducted in 2018 and 2019 in a 54-hectare organic pear orchard in Daxing district, Beijing, China (116.45° E, 39.98° N). The majority of trees were *Pyrus pyrifolia* (cv. Cuiyu), five years old with an average height of 2.5 m, planted ca. 0.7 m apart with 3.3 m row spacing. Prior to the study, the orchard had a history of losses due to fruit damage by *G. molesta*. The orchard was divided into four main parts by two roads running across it: east-west and north-south. The southwest quadrant was divided into three plots, each separated by a buffer of 15 m, and intercropped with either natural grass cover, okra, or castor bean (Figure 1). The intercrops were planted in mid-April and were watered and weeded weekly. For each treatment, five blocks of pear trees (330 m^2^ each) were established as experimental replicates, and the flight activity of male *G. molesta* moths was monitored from June to October by means of a sex pheromone trap (PHEROBIO, Beijing, China), each attached to a pear tree at a height of 1.5 m at the center of each block, with one trap per section. Although it could be argued that this layout did not provide sufficient treatment replication, we contend that the replication of treatment blocks within plots, together with the buffers between plots, was sufficient to provide adequate interspersion of treatments [30], effectively resolving treatment effects that would not likely have been observed if plot-specific factors had biased results. The traps were checked and sticky sheets replaced weekly, and pheromone lures were replaced monthly, per the manufacturer. The entire orchard was managed conventionally throughout the investigation period, with applications of the pesticides spirotetramat, emamectin benzoate, abamectin, pyridaben, and thiamethoxam applied at petal-fall, post-bloom, fruit set, young fruit, and fruit enlargement stages, respectively, to control a complex of phloem-feeding pests (aphids, psyllids, and scales) and folivores (lepidopterous larvae and mites). 

### 2.2. Volatile Collection and Analysis

Volatiles were collected from fresh samples of leaves, flowers, and fruits of okra and leaves and flowers of castor bean (ca. 30 g each) using a dynamic headspace air collection device (model QC-1, Beijing Municipal Institute of Labor Protection, Beijing, China) as described in previous work [31]. Porapak Q (80/100, 50 mg) tubes were used to absorb odors, and these were connected to an oven bag (Reynolds, 406 mm × 444 mm, Lake Forest, IL, USA) where the volatiles were delivered. Air entering the collection device is first passed through activated charcoal and distilled water, and then through a bag containing the odor source. Volatiles were collected from each sample for 2 h at 25 °C with an airflow rate of 50 mL min^−1^. After collection, the absorbent was eluted three times with hexane in a total volume of 1 mL, and the sample obtained was analyzed using GC/MS (Agilent 7890B&7200, Agilent, Santa Clara, CA, USA). One microliter of each sample was injected into a 200 °C injector with a column temperature of 40 °C for 3 min, and the temperature was increased to 270 °C at a rate of 6 °C min^−1^. Compounds were identified by comparing the mass spectra against the NIST 11 Mass Spectral Library (NIST/2011/EPA/NIH), and the relative ratio of each compound in samples was calculated accordingly using the area normalization method. 

### 2.3. Insect Rearing

Larvae of *G. molesta* (ca. 300) were collected from pear orchards in the Institute of Pomology in Liaoning province, China (120.12° E, 40.1° N) and used to establish a laboratory colony. Larvae were reared on Fuji apple fruits for more than thirty generations under constant conditions of 25 ± 1 °C, 70 ± 5% RH, and a 16:8 (L:D) photoperiod [32]. Briefly, pupae were sexed and placed in ventilated cylindrical plastic containers (15.0 cm diam × 9.0 cm ht), 40 per container (sex ratio = 1:1). Upon emergence, adults were provided with a 10% honey solution on balls of cotton (replaced daily), and females laid eggs on the inner surface of the container. Eggs were harvested daily by moving adults to new containers and cutting the containers into pieces to produce cards bearing ca. 20 eggs each. Twenty cards were then placed with 10 apples in each of a series of plastic baskets (50 × 30 × 10 cm), so that eclosing neonates could bore into the fruit directly. Larvae emerged from the fruit as mature fifth-instars and pupated in layers of gauze placed on the top and bottom of each basket.

A laboratory colony of *T. dendrolimi* was established with wasps purchased from Gongzhuling Jinong Green Agriculture High-tech Development Co., Ltd., Jilin, China, and reared on eggs of *Antheraea pernyi* (Guerin-Meneville) (Lepidoptera: Saturniidae) as a factitious host. Female *T. dendrolimi* used in olfactometry were reared on eggs of *G. molesta* for one generation at 25 ± 1 °C, 70 ± 5% RH, and a 14:10 (L:D) photoperiod [15]. Briefly, one d-old *T. dendrolimi* females (n = 100) were confined with males (n = 100) in plastic Petri dishes (9 cm diam) for 24 to permit mating. Wasps were provisioned with a 10% honey solution smeared on the Petri dish covers. Host eggs were parasitized by confining female mated wasps for 24 h in plastic Petri dishes (n = 2 wasps per dish), each containing a *G. molesta* egg card with ca. 50 eggs.

### 2.4. Olfactometry

In order to test the responses of *G. molesta* adults and female *T. dendrolimi* adults to volatiles emanating from okra and castor bean, choice tests were conducted in a Y-tube olfactometer (2.0 cm internal diam) with a stem length of 10 cm and arms 8 cm long, with a 75° angle at the branch point. Five volatile organic compounds were selected as odor sources based on their identification as constituents of okra headspace (*α*-copaene, thymol, and 6-methylhept-5-en-2-one) and castor bean headspace (cinnamaldehyde and dibutyl phthalate) and their commercial availability. Among these, *α*-copaene is known to be an effective attractant for various herbivores [33,34], whereas cinnamaldehyde has both repellent and insecticidal activity [35,36,37]. All compounds were purchased from Sigma-Aldrich Corp. with a purity of 98.0–99.5% and tested at concentrations of 5%, 2%, and 1% using ethanol as a carrier solvent. A 10 μL droplet of each compound in solution was placed on a piece of filter paper (8 cm^2^), and individual samples were placed inside a ground glass stoppered flask (4 cm diameter, 10 cm height); filter paper with a droplet of pure ethanol served as a control. Air was drawn through the apparatus by means of a vacuum pump (as above) placed upstream of the release chamber to carry the volatile compounds through the arms of the olfactometer. The air was first purified by passage through activated charcoal, humidified by passage through distilled water, and entered each odor source bottle at a flow rate of 500 mL min^−1^. Tests were carried out at 25.0 °C under a red light placed directly above the apparatus. Bioassays were conducted between 19:00 and 22:00 h for *G. molesta*, the typical period of moth activity, and between 9:00 and 16:00 h for *T. dendrolimi*. Each moth or female parasitoid was introduced singly at the entrance of the main arm and walked upwind to choose between the two odor sources. Choices were recorded when an insect walked >2 cm beyond the branch point or reached the end of one arm; otherwise, no choice was recorded after 10 min of observation. The Y-tube was reversed after each observation, all tubes were rinsed with 95% ethanol before reuse, and the Y-tube was replaced after every ten observations. 

### 2.5. Statistical Analysis

Statistical analyses were all performed using the SPSS Statistics package ver. 17.0 [38]. Total trap catches of *G. molesta* were compared among intercrop treatments by one-way ANOVA followed by Tukey’s test after square-root transformation of the data with the formula of $\sqrt{x+0.5}$, where *x* represents the catch number [39]. Insect responses to different odor sources were compared by chi-square. 

## 3. Results

### 3.1. Intercrop Experiment 

Two peaks of *G. molesta* activity were observed in 2018, a small peak in early August and a larger one in early September (Figure 2A), whereas in 2019, there were two small peaks in July and two larger ones in late August/early September (Figure 2B). Total catches of *G. molesta* males in 2018 were significantly lower in the okra intercrop (mean ± SE = 110.4 ± 8.4) than in either the grass cover (mean ± SE = 179.4 ± 8.9) or castor bean (mean ± SE = 165.6 ± 8.8) treatments (*F*_2,4_ = 17.40, *p* < 0.001), and in 2019, catches were lower in both the okra (mean ± SE = 88.4 ± 18.5) and castor bean (mean ± SE = 111.0 ± 11.0) treatments than in the grass cover control (mean ± SE = 182.6 ± 26.4) (*F*_2,4_ = 9.07, *p* = 0.004).

### 3.2. Volatile Analysis

Six principle types of volatile compounds were collected from okra and castor bean: hydrocarbons, alcohols, phenols, aldehydes, ketones, and esters (Appendix A). A higher diversity of volatiles and comparatively more hydrocarbons emanated from okra, whereas castor bean released more aldehydes, ketones, and esters (Figure 3).

### 3.3. Olfactometry of Grapholita molesta

Five commercially available volatile compounds, identified from okra (thymol, 6-methylhept-5-en-2-one, and α-copaene) and castor bean (cinnamaldehyde and dibutyl phthalate), were tested against clean air for attraction/repulsion of *G. molesta* moths in an olfactometer. Negative responses to most compounds were observed, although results were variable across concentrations (Figure 4). Compared with the three okra volatiles, the two compounds from castor showed more obvious repellency to both male and female *G. molesta*, especially cinnamaldehyde (Figure 4D).

### 3.4. Olfactometry of Trichogramma dendrolimi Females

Similar to the response of *G. molesta*, female *T. dendrolimi* expressed negative responses to the three volatiles of okra and one volatile (dibutyl phthalate) from caster bean (Figure 5). However, cinnamaldehyde exhibited a significant attractive effect when diluted 50-fold (*χ*^2^ = 21.87, *p* < 0.001; Figure 5D). 

## 4. Discussion

Results of the orchard trial revealed that both okra and castor bean intercrops could reduce the density of *G. molesta* moths recruited to the pear orchard, although the effect of castor bean was significant only in the second year of observations and actually led to higher moth densities during the period of peak flight in the first year. However, in both years, the number of *G. molesta* trapped in the okra intercrop treatment was significantly lower than the number trapped in the grass cover treatment. Previous work in the same orchard has shown that trap catches of *G. molesta* provide a good estimate of levels of subsequent fruit damage [15]. 

Volatile organic compounds collected from okra were predominantly hydrocarbons, phenols, and ketones, whereas the headspace of castor bean was rich in aldehydes, ketones, and esters. Certain volatile hydrocarbons emitted by plants can be repellent to herbivorous insects [40]. Although many benzenoid compounds were identified in the headspace of okra, their potential to influence herbivore behavior remains unexplored to date. Certain ketones, such as the primary component in the essential oil of *Artemisia annua*, exhibit antimicrobial activity and repellency to storage pests [41]. Another originally extracted from wild tomato plants, methyl nonyl ketone (also known as 2-undecanone), exhibits repellency toward ticks [42]. Of the ketones identified in the present study, analogues of *β*-damascone, identified from castor bean, have been shown to inhibit aphid feeding behavior [43]. On the other hand, the alcohols, aldehydes, and esters that are abundant in castor bean volatiles are often associated with the attraction of insects to plants [40]. Methyl jasmonate, an important signaling molecule triggering plant defensive reactions, was detected in both okra and castor bean, with higher amounts emanating from okra. Despite some uncertainty with respect to the roles of specific compounds, we can infer that the volatile emissions of both okra and castor bean were likely responsible for the observed reductions in *G. molesta* trap catches in sections of the pear orchard where these plants were intercropped. It is also possible that the presence of intercrops increased biodiversity by attracting additional insect species, including arthropod predators, that may have deterred moth recruitment via ‘ecology of fear’ effects [44]. 

Based on the GC-MS results, five commercially available volatile organic compounds were tested for olfactory repellency to *G. molesta* and its egg parasitoid, *T. dendrolimi*. Some level of repellency to *G. molesta* was observed for all five compounds, but particularly for cinnamaldehyde, dibutyl phthalate, and thymol. Cinnamaldehyde, a component of castor bean volatiles, exhibits antibacterial activity [45] and has been studied in nanoencapsulated form as a potential mosquito repellent when applied to fabrics [46] and as a repellent to the Indian meal moth, *Plodia interpunctella* (Hübner) (Lepidoptera: Pyralidae), when incorporated into packaging material for stored products [47]. It has also shown repellency toward weevil pests in stored products [48,49], insecticidal activity against *Spodoptera littoralis* (Boisduval) (Lepidoptera: Noctuidae) [50] and *Musca domestica* L. (Diptera: Muscidae) [37], and feeding inhibition in *Drosophila suzukii* (Matsumura) (Diptera: Drosophilidae) [51]. In our study, cinnamaldehyde was deterrent to both male and female *G. molesta* (although not for males at 50-fold dilution) but exhibited no deterrency to *T. dendrolimi* females, even serving to attract them in the 50-fold dilution trial. Thus, cinnamaldehyde emerged as the most promising compound for potential deployment in management programs for *G. molesta* in pear orchards. Given that the primary source of cinnamaldehyde is cinnamon itself, a widely used and valuable cooking spice, *Cinnamomum* spp. could merit investigation as an intercrop with cash value in addition to repellent properties. 

Dibutyl phthalate, a primary component of castor bean volatiles, exhibited repellency to *G. molesta* moths, particularly when diluted 20-fold, although it was also repellent to *T. dendrolimi*. Dibutyl phthalate is an endocrine-disrupting chemical causing various mammalian pathologies [52] that has acaricidal activity against dust mites [53], but there has been no examination to date of any possible effects on insect behavior. Of the three volatile constituents identified from okra, *α*-copaene was repellent to male *G. molesta* at 50-fold dilution, repellent to females at 100-fold dilution, and lacked repellency to the egg parasitoid at these concentrations. This compound has been identified as one of several sesqueterpenes released by chili peppers in response to feeding by the aphid *Myzus persicae* (Sulzer) (Hemiptera: Aphididae) that causes aphid-infested plants to become less attractive to the whitefly *B. tabaci* [54]. Although thymol was repellent to male moths at all tested dilution rates and to *T. dendrolimi* at the 20-fold rate, it did not repel females and even attracted them at the 50-fold rate. Similarly, 6-methylhept-5-en-2-one repelled male *G. molesta* at 50- and 100-fold dilutions and *T. dendrolimi* at 50-fold, but had no measurable effect on female moths. Thymol has been implicated in synergizing the antimicrobial effect of nicotine on the trypanosome gut parasitoid *Crithidia bombi*, Lipa, and Triggiani (Kinetoplastea: Trypanosomatidae) when supplied in the diet of the bumble bee *Bombus impatiens* [55]. 

Given that our behavioral assays were limited to the five volatile components that were commercially available, we cannot overlook the potential importance of other volatile constituents to the inhibitory effect of these intercrops on the recruitment of *G. molesta* to the pear orchard. It is also possible that the reduced trap catches of moths observed in the intercropped orchard plots result from net responses to combinations of volatiles or interactions (antagonisms and synergisms) between constituents and thus may not be attributable to any single compound. Furthermore, if the intercrops altered the composition of the local arthropod community, the increased insect diversity could have generated non-consumptive, enemy-risk effects that contributed to *G. molesta* deterrence.

Okra and castor beans offer other potential benefits to orchard pest management that were not assessed in the present study. In addition to their repellency to *G. molesta* adults, the presence of these plants as intercrops may contribute to arthropod diversity in the orchard, thus promoting biological control of other pests. For example, okra can host leaf beetles, cotton aphids, and whiteflies that will support populations of generalist predators such as spiders, syrphids, coccinellids, and lacewings [19], which may then contribute to the control of aphids, psyllids, and gall midges on the pear trees. Castor bean has allelopathic properties that may assist with suppression of weed species [56] and has been successfully employed as a banker plant to enhance populations of the whitefly parasitoid *Encarsia formosa* Gahan (Hymenoptera: Aphelinidae) for control of *B. tabaci* in tomato [57]; feeding by whiteflies on *R. communis* elicits the production of volatiles that attract the parasitoid [58]. Therefore, more robust assessments of the impact of these intercrops on the arthropod community in pear orchards are warranted to identify potentially broader ecological benefits. These should include estimates of the degree to which fruit infestation can be reduced by okra and caster bean intercrops or by the application of their specific volatile components. 

## 5. Conclusions

We conclude that okra and castor bean emit volatiles that are repellent to *G. molesta* adults and can reduce the recruitment of moths when these plants are intercropped in pear orchards (Figure 6). Given that okra and castor bean host various herbivorous insects that serve as prey for generalist arthropod predators contributing to biological pest control within the orchard ecosystem, they may also serve as banker plants that further enhance integrated pest management programs in pear orchards. Among the plant volatiles evaluated, cinnamaldehyde, either directly applied or emanating from intercropped *Cinammomum* spp., could serve to repel *G. molesta* without disrupting augmentative releases of *T. dendrolimi*. 

## Figures and Tables

**Figure 1 insects-14-00885-f001:**
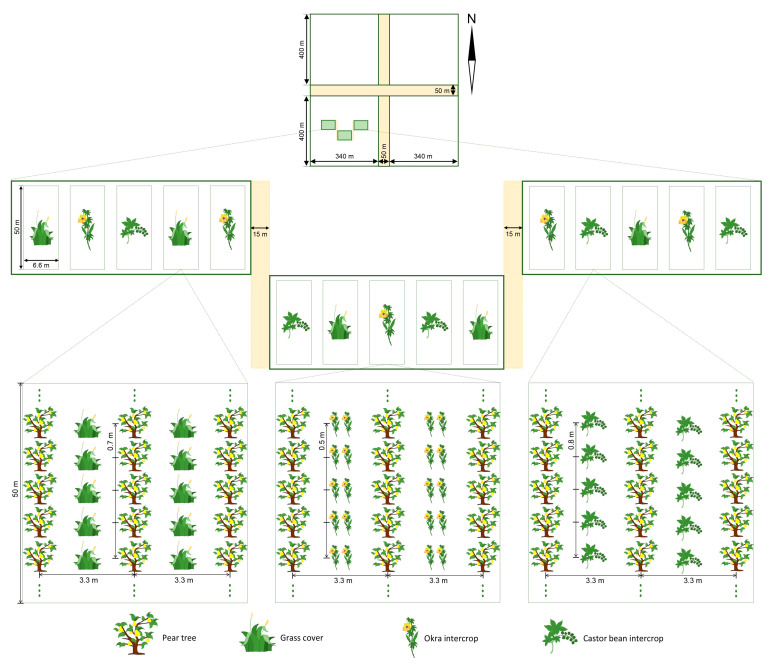
A schematic diagram of the three regions of the pear orchard depicts the placement of the natural grass cover, okra, and castor bean intercrop treatments.

**Figure 2 insects-14-00885-f002:**
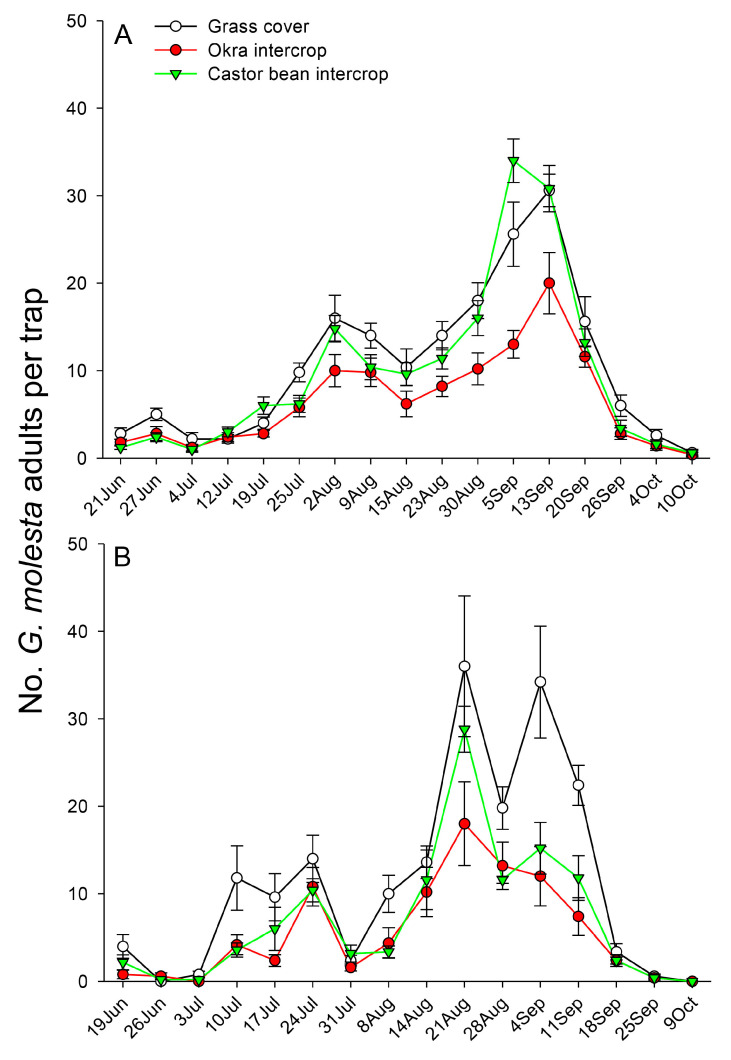
Mean (±SE) number of *Grapholita molesta* males caught per trap in a pear orchard intercropped with three different plants (**A**) in 2018 and (**B**) in 2019.

**Figure 3 insects-14-00885-f003:**
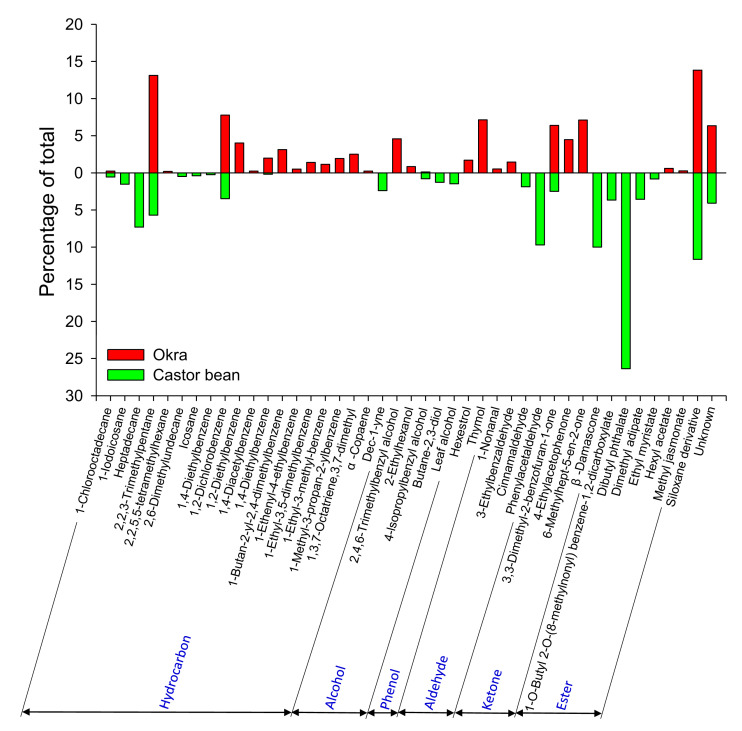
Relative proportions of volatile organic compounds identified in headspace from okra and castor bean as determined by GC/MS analysis.

**Figure 4 insects-14-00885-f004:**
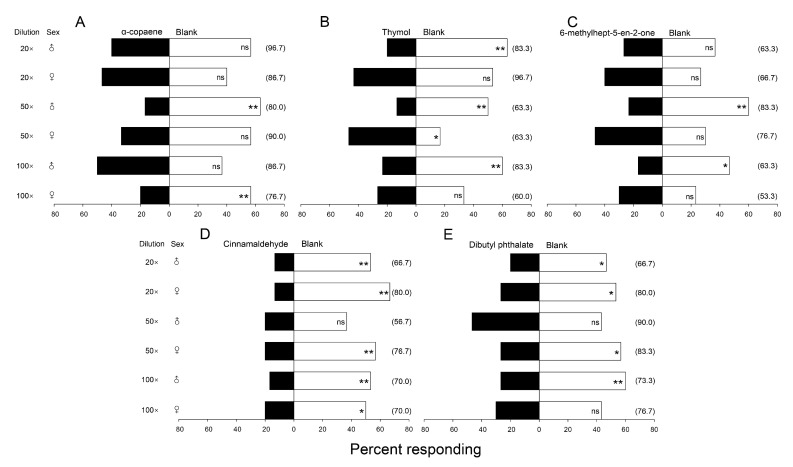
Percentages of male and female *Grapholita molesta* adults (*n* = 30 moths per trial, tested singly) responding to three volatiles from okra: *α*-copaene (**A**), thymol (**B**), and 6-methylhept-5-en-2-one (**C**), and two volatiles from castor bean: cinnamaldehyde (**D**) and dibutyl phthalate (**E**), when presented versus clean air in a Y-tube olfactometer (chi-square test, *: *p* ≤ 0.05, **: *p* ≤ 0.01). Each compound was tested at three dilutions; numbers in parentheses indicate the percentage of moths responding in each test.

**Figure 5 insects-14-00885-f005:**
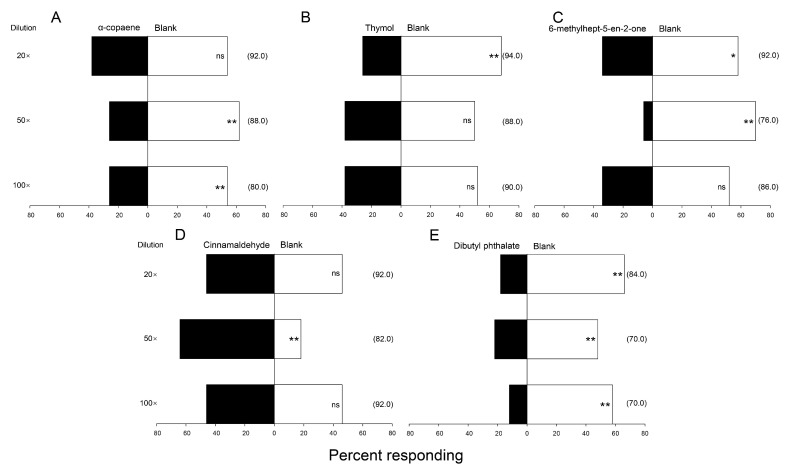
Percentages of *Trichogramma dendrolimi* females (n = 50 wasps per trial, tested singly) responding to three volatiles from okra: *α*-copaene (**A**), thymol (**B**), and 6-methylhept-5-en-2-one (**C**), and two volatiles from castor bean: cinnamaldehyde (**D**) and dibutyl phthalate (**E**), when presented versus clean air in a Y-tube olfactometer (chi-square test, *: *p* ≤ 0.05, **: *p* ≤ 0.01). Each compound was tested at three dilutions; numbers in parentheses indicate the percentage of wasps responding in each test.

**Figure 6 insects-14-00885-f006:**
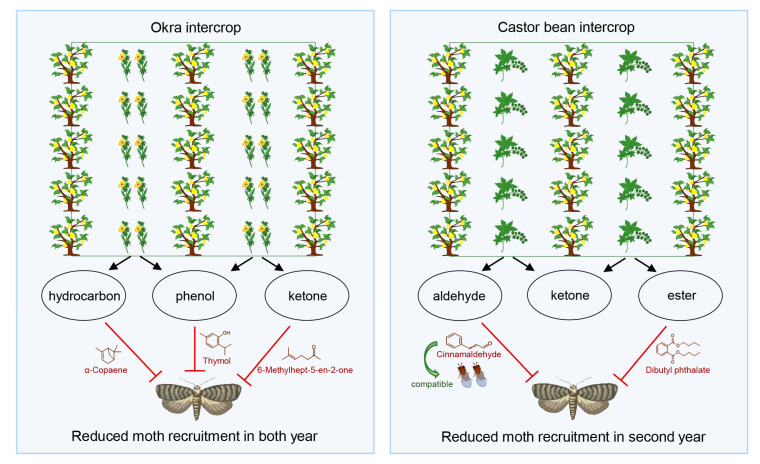
Schematic diagram of hypothesized effects of okra and castor intercropping on population control of *Grapholita molesta* in a pear orchard.

## Data Availability

The data presented in this study are available on request from the first author (Z.L).

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
