# Peer review of "Intercropping Okra and Castor Bean Reduces Recruitment of Oriental Fruit Moth, Grapholita molesta (Lepidoptera: Tortricidae) in a Pear Orchard"

_insects, 2023, doi:10.3390/insects14110885_

Round 1

Reviewer 1 Report

Comments and Suggestions for Authors

Dear Authors,

You have carried out a study that may have great potential from a scientific and practical point of view, but unfortunately you have not carried it out in full.

I have some doubts about the conduct of the experiment. You write that the experiment was conducted in 2018 and 2019 in a 54 hectare organic pear orchard (line 103), afterwards you state that the entire orchard was managed conventionally throughout the period of the investigation (lines 116-117).

Furthermore, the fact that volatile organic compounds can decrease the number of male catches in pheromone traps has little practical significance in terms of safeguarding orchard production. Moreover, there is great variability in the capture of males in pheromone traps, which is why you should have increased the number of traps.

To have data of scientific and practical value, you should have verified the infestation on the fruit. 

You should therefore have additional data that take the infestation into account. Alternatively, you could eliminate the part of the experiment that you carried out in the field, but the work would be greatly impaired.

Specific comments:

Line 110: The three sections into which the pear orchard was divided were located in which part of the pear orchard? Need to include a map of the location of the sections where the tests were conducted.

Lines 114-116: How often were the traps checked? Was the pheromone replaced during the monitoring period? How often?

The materials and methods must be more detailed.

 The manuscript presented in its current form cannot be accepted.

Author Response

General Comments

You have carried out a study that may have great potential from a scientific and practical point of view, but unfortunately you have not carried it out in full.

I have some doubts about the conduct of the experiment. You write that the experiment was conducted in 2018 and 2019 in a 54 hectare organic pear orchard (line 103), afterwards you state that the entire orchard was managed conventionally throughout the period of the investigation (lines 116-117).

Furthermore, the fact that volatile organic compounds can decrease the number of male catches in pheromone traps has little practical significance in terms of safeguarding orchard production. Moreover, there is great variability in the capture of males in pheromone traps, which is why you should have increased the number of traps.

To have data of scientific and practical value, you should have verified the infestation on the fruit. 

You should therefore have additional data that take the infestation into account. Alternatively, you could eliminate the part of the experiment that you carried out in the field, but the work would be greatly impaired.

RESPONSE:

The objective of our study was to generate novel insights that would improve management of the oriental fruit moth. A first step was to demonstrate repellency of okra and castor bean to G. molesta in a field setting (a pear orchard). We further explored the underlying mechanism of repellency to obtain information that might improve management of G. molesta and other orchard pests. We agree that additional traps would have improved resolution of the data, as would sampling infested fruit, but sampling efforts are always limited by available manpower and practical feasibility. However, males orient to female sex pheromones, and both mating disruption and mass-trapping tactics have effectively employed these pheromones, so it follows that male presence will correlate with female presence, and that catches of male moths would not likely decline in treatment blocks unless fewer females were present. However, we have acknowledged this limitation of the present study on line 334 of the discussion.

Specific Comments

Line 110: The three sections into which the pear orchard was divided were located in which part of the pear orchard? Need to include a map of the location of the sections where the tests were conducted.

RESPONSE: We have modified Fig.1. to better depict placement of the three intercrop treatments. The verbal description provided in the section 2.1 has also been clarified (lines 110-114).

Lines 114-116: How often were the traps checked? Was the pheromone replaced during the monitoring period? How often? The materials and methods must be more detailed.

RESPONSE: The requested information has been added on lines 117-119 of M&M.

Reviewer 2 Report

Comments and Suggestions for Authors

MS entitled “Intercropping Okra and Castor Bean Reduces Recruitment of Oriental Fruit Moth, Grapholita molesta (Lepidoptera: Tortricidae) in a Pear Orchard” is devoted to an interesting and important issue in theoretical and applied terms. It is well known that intercropping is a tactic of cultural control, which is of interest from the point of view of increasing biodiversity and improving chemical communication between plants and insects in agroecosystems. At the same time, it should be noted that the effect of intercropping has not yet been sufficiently studied, and therefore the material presented in the reviewed MS is of undoubted interest. First, data obtained by the authors demonstrate that intercropping okra and castor bean in a pear orchard reduce pheromone trap catches of Grapholita molesta. Second, GC-MS assays proved that okra and castor bean emit some volatiles that are repellent to G. molesta adults. It is also very interesting to note that one of these volatiles, namely cinnamaldehyde, repels G. molesta adults but could exhibit an attractive effect to Trichogramma dendrolimi females.

It seems however that the MS contains quite strong flaws which must be taken into account to improve the paper. So, in the Materials and Methods section, the authors write “The orchard was divided into three sections, each separated by a buffer of 15 m, and each section was intercropped with either natural grass cover, okra, or castor bean. The intercrops were planted in mid-April and were watered and weeded weekly. Within each treatment section, five blocks of pear trees (330 m² each), were established as experimental replicates and the flight activity of male G. molesta moths was monitored from June to October by means of a sex pheromone trap attached to pear tree at a height of 1.5 m at the center of each block”. (lines 110-116). The text presented above gives grounds to assume that the block organization of the experiments was the so-called pseudorepetitions (see: Hurlbert, S.H. (1984). Pseudoreplication and the design of ecological field experiments. Ecological monographs, 54(2): 187-211. https://doi.org/10.2307/1942661).

Further, section 2.5. Statistical analysis reports us: “Total trap catches of G. molesta were compared among intercrop treatments by one way ANOVA followed by Fisher's LSD test (lines 194-195). It should be noted that, as a rule, catches of lepidopteran insects in pheromone traps are characterized by significant deviations from the normal distribution, and therefore it is often recommended to use data transformation (for example, square-root transformation) (see: Roelofs, W. L., Cardé, R. T. (1977). Responses of Lepidoptera to synthetic sex pheromone chemicals and their analogues. Annual Review of Entomology, 22: 377–405. https://doi.org/10.1146/annurev.en.22.010177.002113). Anyway, the question of the normality of the distribution of butterfly catches in traps should be given attention.

Author Response

General Comments

MS entitled “Intercropping Okra and Castor Bean Reduces Recruitment of Oriental Fruit Moth, Grapholita molesta (Lepidoptera: Tortricidae) in a Pear Orchard” is devoted to an interesting and important issue in theoretical and applied terms. It is well known that intercropping is a tactic of cultural control, which is of interest from the point of view of increasing biodiversity and improving chemical communication between plants and insects in agroecosystems. At the same time, it should be noted that the effect of intercropping has not yet been sufficiently studied, and therefore the material presented in the reviewed MS is of undoubted interest. First, data obtained by the authors demonstrate that intercropping okra and castor bean in a pear orchard reduce pheromone trap catches of Grapholita molesta. Second, GC-MS assays proved that okra and castor bean emit some volatiles that are repellent to G. molesta adults. It is also very interesting to note that one of these volatiles, namely cinnamaldehyde, repels G. molesta adults but could exhibit an attractive effect to Trichogramma dendrolimi females.

Specific Comments

It seems however that the MS contains quite strong flaws which must be taken into account to improve the paper. So, in the Materials and Methods section, the authors write “The orchard was divided into three sections, each separated by a buffer of 15 m, and each section was intercropped with either natural grass cover, okra, or castor bean. The intercrops were planted in mid-April and were watered and weeded weekly. Within each treatment section, five blocks of pear trees (330 m² each), were established as experimental replicates and the flight activity of male G. molesta moths was monitored from June to October by means of a sex pheromone trap attached to pear tree at a height of 1.5 m at the center of each block”. (lines 110-116). The text presented above gives grounds to assume that the block organization of the experiments was the so-called pseudorepetitions (see: Hurlbert, S.H. (1984). Pseudoreplication and the design of ecological field experiments. Ecological monographs, 54(2): 187-211. https://doi.org/10.2307/1942661).

RESPONSE: We have modified Fig.1 and revised the description in the text to clarify the plot arrangement. The reviewer is correct, and it can be argued that the experimental design suffers from pseudoreplication; we have acknowledged this in the M&M, cited the relevant reference, and explained why we believe this would not have compromised the observed treatment effects (lines 118-122).

Further, section 2.5. Statistical analysis reports us: “Total trap catches of G. molesta were compared among intercrop treatments by one way ANOVA followed by Fisher's LSD test (lines 194-195). It should be noted that, as a rule, catches of lepidopteran insects in pheromone traps are characterized by significant deviations from the normal distribution, and therefore it is often recommended to use data transformation (for example, square-root transformation) (see: Roelofs, W. L., Cardé, R. T. (1977). Responses of Lepidoptera to synthetic sex pheromone chemicals and their analogues. Annual Review of Entomology, 22: 377–405. https://doi.org/10.1146/annurev.en.22.010177.002113). Anyway, the question of the normality of the distribution of butterfly catches in traps should be given attention.

RESPONSE: The catch numbers of male G. molesta have been transformed and then statistically analyzed by one way ANOVA followed by Tukey’s test, according to Roelofs, 1997 (now cited) and the description of the analysis has been revised accordingly (lines 201-203). The results of the analysis have been similarly revised in section 3.1 (lines 212-214). 

Round 2

Reviewer 1 Report

Comments and Suggestions for Authors

Dear Authors,

as I wrote to you in my previous comment, the experiment shows some weaknesses. You have carried out this study on an orchard that cannot be considered organically managed. The biological conduct in these studies is of fundamental importance. You use chemical insecticides that can alter the validity of the results on the real population of Tortricidae. Furthermore, a lower capture of male moths in pheromone traps does not often translate into a lower infestation of fruits. As you well write, the male presence is correlated to the female presence but a female who mates, as you know, can lay more eggs, which translates into infestation of the fruits. I therefore reiterate that the analysis of the infestation on the fruits in this work appears decisive. Your response “sampling efforts are always limited by available manpower and practical feasibility” cannot be accepted in such an important and complex study. I suggest resubmitting the manuscript exclusively for the part that concerns the laboratory activity.

The manuscript presented in its current form cannot be accepted.

Author Response

Yes, the application of chemical insecticides could have potentially affected overall tortricid populations, but given that all intercrop treatment blocks were treated exactly the same, the observed treatment effects could not have been the result of any chemical treatment. Therefore, differences in G. molesta populations among treatments had to result from the different intercrops.

Previous work in the same orchard that did assess fruit damage showed that numbers of male moths captured in pheromone traps were correlated with levels of fruit damage (Zhang et al., Pest Management Science, 2021, 77: 2795-2803). We have now included a statement to this effect, along with the supporting citation, on line 257.

We prefer to retain the field test in the manuscript, as it provided the rationale for the laboratory study, and generated complementary results, Whereas the laboratory work illustrates the likely mechanisms responsible for repellency, the field data show that the intercrop plants can influence moth densities locally in the field. An article presenting the laboratory data alone would seem lacking in rationale.

We appreciate the constructive suggestions to improve the manuscript. Please contact us if any further changes are required.

Yours sincerely,

Huanli Xu

Associate professor, Ph.D.

China Agricultural University